# Adolescence, Adulthood and Self-Perceived Halitosis: A Role of Psychological Factors

**DOI:** 10.3390/medicina57060614

**Published:** 2021-06-12

**Authors:** Carmela Mento, Clara Lombardo, Mariacristina Milazzo, Nicholas Ian Whithorn, Montserrat Boronat-Catalá, Pedro J. Almiñana-Pastor, Cristina Sala Fernàndez, Antonio Bruno, Maria Rosaria Anna Muscatello, Rocco Antonio Zoccali

**Affiliations:** 1Department of Biomedical, Dental Sciences and Morphofunctional Imaging, University of Messina, Psychiatric Unit Policlinico “G. Martino” Hospital, 98124 Messina, Italy; csalafernandez@gmail.com (C.S.F.); antonio.bruno@unime.it (A.B.) mmuscatello@unime.it (M.R.A.M.); rocco.zoccali@unime.it (R.A.Z.); 2Psychiatric Unit, Policlinico Hospital “G. Martino”, 98124 Messina, Italy; clara.lombardo1988@gmail.com (C.L.); Mary-cry13@hotmail.it (M.M.); 3Department of Political Sciences, University of Messina, 98122 Messina, Italy; nwhithorn@yahoo.it; 4Department of Stomatology, Faculty of Medicine and Dentistry, University of Valencia, 46010 Valencia, Spain; montse_boronat@hotmail.com (M.B.-C.); pedroalminana@gmail.com (P.J.A.-P.)

**Keywords:** self-perceived halitosis, psychological factors, breath odor, halitosis and social relationship

## Abstract

(1) Background: Halitosis is a frequent condition that affects a large part of the population. It is considered a “social stigma”, as it can determine a number of psychological and relationship consequences that affect people’s lives. The purpose of this review is to examine the role of psychological factors in the condition of self-perceived halitosis in adolescent subjects and adulthood. (2) Type of studies reviewed: We conducted, by the PRISMA (Preferred Reporting Items for Systematic Review and Meta-Analyses) guidelines, systematic research of the literature on PubMed and Scholar. The key terms used were halitosis, halitosis self-perception, psychological factors, breath odor and two terms related to socio-relational consequences (“Halitosis and Social Relationship” OR “Social Issue of Halitosis”). Initial research identified 3008 articles. As a result of the inclusion and exclusion criteria, the number of publications was reduced to 38. (3) Results: According to the literature examined, halitosis is a condition that is rarely self-perceived. In general, women have a greater ability to recognize it than men. Several factors can affect the perception of the dental condition, such as socioeconomic status, emotional state and body image. (4) Conclusion and practical implication: Self-perceived halitosis could have a significant impact on the patient’s quality of life. Among the most frequent consequences are found anxiety, reduced levels of self-esteem, misinterpretation of other people’s attitudes and embarrassment and relational discomfort that often result in social isolation.

## 1. Introduction

Halitosis, commonly called bad breath, is a problem that can affect both the external, relational, social communication and internal, psychological sphere, with implications on perceived quality of life [1]. People are usually unaware of their breath, and when they become aware of it, they incorrectly attribute the cause of their condition. Halitosis is a frequent condition, present in 50–65% of the world population [2]. Although it is a significant source of discomfort, a precise estimation of prevalence is not possible because epidemiological studies are limited. This limitation could be determined by several factors, such as the absence of a standardized method for assessment of the disease, the difficulty in recognizing its presence and the likelihood that it is in some cases transient, which is why it is underreported in epidemiological data [3]. The problem of people with halitosis is that this condition can often remain unnoticed because people are generally unaware of the quality of their oral odor. Research conducted on Korean adolescent subjects [4] has made it possible to identify a considerable number of adolescents between 12 and 18 years of age with halitosis, highlighting the contributing factors at this age such as substance abuse and smoking, diet and economic social status. Regarding gender, there appears to be no correlation whatsoever [5,6]. The term physiological or transient halitosis refers to dental bad smell, which occurs only at certain times of the day, for example during the early morning hours. Occurring in the absence of a specific dental pathology, it can be associated with factors such as diet and tobacco use [3]; psychiatric illnesses, anxiety, depression and stress are also among the causes of halitosis [4,5,6,7]. Halitophobia or pseudohalitosis refers to the fear of suffering from bad breath. In such subjects, treatments aimed at resolving the bad smell would be useless, and evaluation and intervention of a psychologist [8] are considered appropriate. Pathological halitosis is divided by some authors into intraoral (90%) and extraoral (10%) halitosis [1]. The back of the tongue is a potential reservoir of bacteria and source of foul-smelling gases; therefore, daily washing should be performed to reduce the number of bacteria and the processes leading to the manifestation of foul smells [9,10,11,12]. Extraoral factors not related to the oral cavity include intake of maltogenic foods (such as garlic, onions, fatty foods, sugars and sweets, which can promote the development of caries and the production of substances with an unpleasant smell), smoking and alcohol, coffee abuse, metabolic disorders (such as liver failure, diabetic ketoacidosis, cirrhosis, renal failure, hiatal hernia), upper respiratory tract disorders (chronic sinusitis, nasal obstruction, nasopharyngeal abscess) and lower respiratory tract disorders (i.e., bronchitis, pulmonary abscess, lung cancer) [3,4,5,6,7,8,9,10,11,12,13]. Studies in the literature allow us to identify how halitosis has a significant impact on the life of the individual. Among the first consequences, it can cause embarrassment and depression [14,15]. In adolescence, it is a significant condition that compromises social development [16]. Considered by Kolo [5] as a “social stigma”, it could become an obsession that dominates a person’s life, determining the onset of factors such as anxiety and psychosocial stress. The hypochondria of some individuals can, in certain cases, determine the onset of what is called “delusional halitosis”, caused by an incorrect assessment of their olfactory perceptions. The first factor to be compressed in subjects with halitosis is the communicative act with consequent impairment of professional interactions [17]. This could negatively affect self-esteem and confidence, reducing quality of life to the point of complete isolation [18]. Awareness of halitosis leads the individual to experience the condition negatively so that it leads to halitophobia, a term used to indicate a condition characterized by excessive concern with the belief of having halitosis [19]. One theme of studies on self-perceived halitosis shows that the problem is often not self-perceived [20,21]. Self-reported halitosis tends to be underestimated mainly because individuals may have difficulty in detecting their own smell or feel embarrassed to expose themselves, in line with the intimate dimension of the relationship with their mouth [22]. Subjects with a good body image pay more attention to their mouth and oral malodor [23]. Moreover, emotional state can also have a negative impact in neglect of investment in one’s own body, also in terms of care and hygiene, and the subject could become more sensitive to bad smell, highlighting a multifactorial psychophysiological problem [24]. In this regard, individuals with psychiatric illnesses, including those with schizophrenia, schizoaffective disorder, depression, and bipolar disorder, may have impaired oral health [25]. For example, the presence of depressive disorder may adversely affect the patient’s hygiene activities and personal care. Depression is a frequent and debilitating disorder characterized by loss of energy, anhedonia, reduced libido and feelings of sadness and despair that interfere with the daily activities of individuals. Such problems will lead to a reduction in the patient’s self-esteem, leading to negative effects on the treatment of mental illness [26]. Moreover, each patient has a specific and different image of “breath smell”: some subjects experience halitosis but the bad smell is neither offensive nor evident, and many authors, to highlight this condition, speak of “halitosis paradox”; i.e., while many develop wrong perceptions about their bad breath, others are not aware [23]. People who have a bad oral odor problem tend not to be aware of their bad breath, while those who do not have a bad breath problem worry excessively about having a bad odor problem. Young and middle-aged people tend to be more alert and anxious about their health, obtaining significantly higher scores in the OLT (organoleptic test, the gold standard to detect bad oral smell) and showing significant symptoms of anxiety and depression, with the anxiety itself increasing oral levels of volatile sulfur compounds (VSCs) [20,21,22,23,24,25,26,27]. A gender difference in the perception of halitosis was highlighted in one study [23]: 21.7% were male and 35.3% were female. Ashwath et al. [23] stated that self-perceived halitosis was more present in dental students. In addition, other significant relationships are found between self-perceived halitosis and other variables, including paternal and maternal education (illiteracy scores high compared to fathers and mothers with a diploma or university degree), dental brushing (the highest scores were related to an absence of brushing, compared to those who practiced oral hygiene three times a day) and the use of dental floss and mouthwash. Recent studies have suggested that socioeconomic inequality may affect halitosis awareness [28] and that halitosis reporters tend to have difficulty contacting the dentist [29]. A further factor to consider is the relationship between smoking and halitosis awareness: 2% of females and 14% of males report smoking, and this has been found to be significantly associated with self-perception. Gender differences in tobacco use have been attributed to cultural issues [30]. Recent studies confirm that self-perception of halitosis may be related to a psychogenic or psychosomatic disorder and has a strong psychological impact [31,32].

## 2. Materials and Methods

A review strategy has been conducted to summarize the available results of experimental studies. This review was conducted according to Preferred Reporting Items for Systematic Reviews and Meta-Analyses (PRISMA) [33] (Figure 1). 

### 2.1. Criteria for Eligibility and Research Strategy

To identify the studies, we performed a systematic literature search on the PubMed, Scholar and ScienceDirect databases. The search was limited to studies written in English. We identified the literature from January 2010 to January 2020 using four key terms related to self-perceived halitosis, namely halitosis, self-perception halitosis, psychological factors and breath odor, and two terms related to sociorelational consequences (“Halitosis and Social Relationship” OR “ Social Issue of Halitosis”). The articles of 1994 and 2001 were only cited to introduce the body image theory and to link it to the perception that each individual has of his or her own smell. The electronic research strategy used is described in Table 1.

Articles were selected online in relation to the title and abstracts; articles were read in full when titles and abstracts were consistent with the objective of our study. Following this procedure, we found 889 articles on the PubMed database, 1950 articles on Scholar and 169 articles on ScienceDirect; after applying the inclusion and exclusion criteria, the total number of relevant publications was reduced to 38.

### 2.2. Study Selection

The articles were included in the review according to the following inclusion criteria: English language, publication in peer-reviewed scientific journals and quantitative information about self-perceived halitosis and oral hygiene. Articles were excluded based on title and abstract screen; review articles, editorial comments and case reports were also excluded. The quality of included studies was appraised by separate methods for qualitative and quantitative studies; features of study design, methodology and analysis were assessed. Qualitative studies were appraised using Critical Appraisal Skills Programme tool (CASP), while the Effective Public Health Practice Project (EPHPP) tool was used for quantitative studies. The search in the PubMed, Scholar and ScienceDirect databases provided a total of 3008 articles; no further studies meeting the inclusion criteria were identified. After eliminating duplicates, further studies were excluded according to the inclusion and exclusion criteria. After screening, 38 studies were selected as appropriate for the present review.

### 2.3. Bias Risk among Studies

In all studies included in this review, a potential bias of the database should be considered. Only articles in English have been used, which could have compromised access to articles published in other languages.

## 3. Results

According to the literature examined, halitosis is a condition that is rarely self-perceived. In particular, 10 articles focus on the self-perception of halitosis, 10 deal with halitosis by defining it and analyzing the epidemiology, 1 study focuses on body image related to the perception of one’s smell, 7 studies analyze the different psychological factors related to halitosis and 10 studies address the sociorelational problems caused by halitosis. Quality assessment of the selected works was performed using CASP and EPHPP tools; the results of quality assessment are reported in Table 2 and Table 3. Most of the qualitative studies were found to have good quality, while some of the quantitative works showed relevant methodological weaknesses such as selection bias, unsatisfactory design and presence of confounders. Despite these limitations, studies made it possible to understand the importance of this condition and the influence it exerts on the psychorelational status of affected subjects. A summary of the selected studies is reported in Table 4. Additionally, references to the selected articles were examined in order to identify further studies that could meet inclusion criteria (Table 5). Studies found that the prevalence of self-perceived halitosis was 22.8% among the participants. The majority of subjects with self-perceived halitosis experienced bad breath on awakening (83.5%). Self-perceived halitosis was more prevalent among males than females, whereas no statistically significant differences were found between age groups. A statistically significant relationship was found between self-perceived halitosis and mouth cleaning time, and a statistically significant relationship was found between self-perceived halitosis and mouth cleaning time, shisha use or cigarette smoking [16]. In this study, it was found that the more people perceived their oral odor, the more likely they were to maintain a distance. A subgroup of individuals was identified who reported maintaining a certain distance when meeting other people, despite a self-perceived “fresh” oral odor. Self-perceived halitosis leads people to keep their distance in social interactions. The ability to recognize and perceive one’s oral condition can be influenced by socioeconomic variables such as age, gender and level of education [31]. There are frequent cases of oral and dental problems in patients with depression, anxiety or schizophrenia [26]. It has been possible to understand how people have difficulty recognizing the presence of halitosis and have limited knowledge about this condition [11]. It is clear that halitosis is an unpleasant symptom and can create relational difficulties with consequences for the individual’s quality of life [3]. The participants in the study were healthy subjects. Participants in some studies underwent organoleptic tests to measure breathing odor and specific questionnaires were given. Some patients with halitosis were compared with healthy subjects.

This study has identified the importance of the consequences that the perception of bad breath has on the psychorelational side and, in general, on patients’ quality of life.

## 4. Discussion

The studies analyzed focus on the importance of self-perception of halitosis and the consequences it has on the life of the individual. It is an oral health problem that could be prevented through appropriate oral hygiene practices and to which attention should be paid, given the influence it has on patients’ quality of life. The subject’s awareness of his or her condition leads to such anxiety that he or she obsessively uses appropriate instruments to eclipse the bad smell, such as sprays, chewing gum, pills or mouthwashes [38]. Although it is a significant condition for the individual, it is not easy to recognize or “self-perceive” it. A study conducted in Saudi Arabia and the United Kingdom [34] showed that 51.9% of the Saudi population and 54.9% of the UK population were aware of their halitosis. In contrast, a study conducted on the population of the Netherlands shows that only 4.2% were able to recognize their bad smell [35]. In individuals with halitosis, certain consequences have a significant impact on the individual. The reduction in self-confidence and self-esteem levels and the onset of depression and stress are the main consequences that can be observed in patients with this condition. The onset of social anxiety is also important, as it could lead the individual to estrangement and avoidance of social interactions. In a study [37], it has been shown that people with halitosis had relationship failures, in particular with individuals of the opposite gender; showed difficulties in achieving their goals; and exhibited compromised performance even in the workplace, considering this condition a “social handicap”. The research obtained also allows us to affirm that, although halitosis causes considerable problems in the subject, it is often not treated but ignored or denied. Patients are not very optimistic about treatment. In the literature, it emerges that the self-perception of halitosis could determine the onset of symptoms such as anxiety, obsessive-compulsive disorders and paranoid ideation, and very often subjects misinterpret the attitude of people who relate to them. There is a need to introduce, together with the professional doctor, the intervention of specialists such as psychologists and psychiatrists to prevent consequences [17,36]. We believe that it is useful to work in a multidisciplinary mode in which one can intervene on both the oral condition and the psychological conditions of the patient to alter inadequate attitudes and wrong beliefs and to encourage the adoption of behaviors that can reduce the bad smell and improve patients’ quality of life.

### Limitations

This study suffers from several limitations: indeed, due to the relatively low number of studies on the topic and their great methodological heterogeneity, a proper formal quality assessment of each work was precluded. Therefore, selection bias was not properly addressed, which can impact the general interpretability of results. In general, care is required when interpreting the results of the cited studies, and further investigations are needed to clarify the relationship between self-perceived and objective halitosis.

## 5. Conclusions

This study has identified the importance of the consequences that the perception of bad breath has on the psychorelational side and, in general, on patients’ quality of life. Literature studies have shown that women have, as a percentage, a greater awareness of their dental condition than men. Socioeconomic status is one of the factors that can influence self-perceived halitosis. Research has also led to an understanding of how body image and emotional state may be a variable that can influence perception. Oral hygiene measures can significantly reduce the bad smell. Oral health education must also be provided in many places, including private dental and medical clinics, and it is important that healthcare professionals, including general practitioners and health professionals, understand the etiology and critical factors in order to diagnose and treat patients appropriately. This review has reviewed the studies in the literature. The limitation of our study concerns the exclusive treatment of psychological factors implicated in halitosis at the expense of possible treatments.

## Figures and Tables

**Figure 1 medicina-57-00614-f001:**
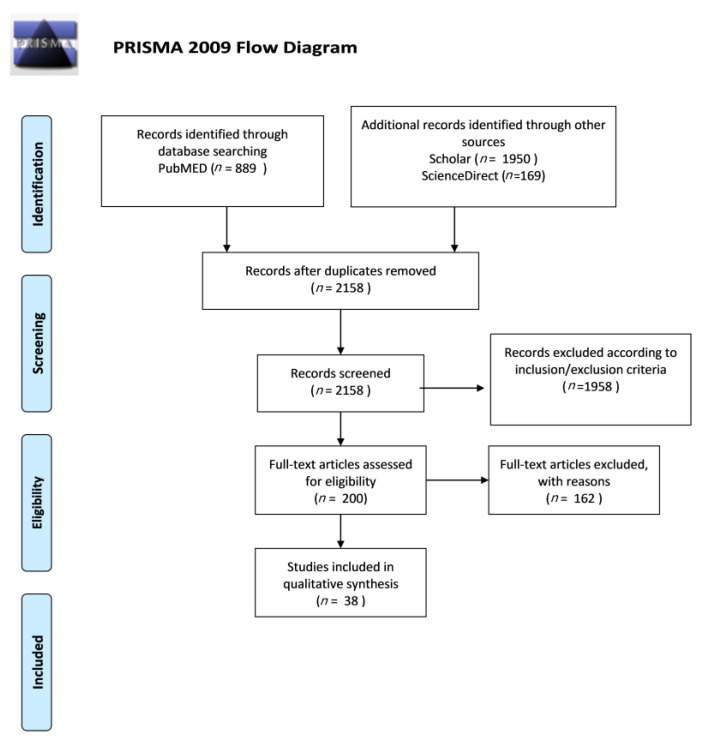
Prisma Flow Diagram.

**Table 1 medicina-57-00614-t001:** List of search terms entered into the PubMed, Scholar and ScienceDirect search.

Number	Search Term
1	Halitosis (all fields)
2	Halitosis self-perception (all fields)
3	Psychological factors
4	Breath odor
5	Halitosis and Social Relationship
	OR
6	Social Issue of Halitosis
7	English (language)
8	2010/01/01 to 2020/01/31 (publication date)

**Table 2 medicina-57-00614-t002:** Critical Appraisal Skills Programme (CASP).

	Dico, G.L. (2018) [20]	Eli, I., et al. (2001) [22]	Heboyan, A., et al. (2019) [17]	Kapoor, U., et al. (2016) [12]	Kisely, S. (2016) [25]	Madhushankari, G.S., et al. (2015) [8]	Ozen, M.E., et al. (2015) [13]	Slade, P.D. (1994) [21]	Torales, J., et al. (2017) [26]
Item 1. Was there a clear statement of the aim of the research?	Y	Y	Y	Y	Y	Y	Y	Y	Y
Item 2. Is a qualitative methodology appropriate?	Y	Y	Y	Y	Y	Y	Y	Y	Y
Item 3. Was the research design appropriate to address the aims of the research?	Y	Y	Y	Y	Y	Y	U	Y	Y
Item 4. Was the recruitment strategy appropriate to the aims of the research?	Y	Y	U	Y	Y	Y	U	Y	Y
Item 5. Was the data collected in a way that addressed the research issue?	Y	Y	U	Y	Y	Y	Y	Y	Y
Item 6. Has the relationship between researcher and participants been adequately considered?	U	U	U	Y	N	U	U	N	U
Item 7. Have ethical issues been taken into consideration?	U	Y	Y	U	N	N	N	N	N
Item 8. Was the data analysis sufficiently rigorous?	Y	Y	U	Y	Y	Y	Y	Y	Y
Item 9. Is there a clear statement of finding?	Y	Y	Y	Y	Y	Y	Y	Y	Y
Item 10. How valuable is the research?	Y	Y	Y	Y	Y	Y	Y	Y	Y
Overall Score	9	9.5	8	9.5	8	8.5	7.5	8	8.5

Y: Yes (1); N: No (0); U: Unclear (0.5).

**Table 3 medicina-57-00614-t003:** Effective Public Health Practice Project (EPHPP).

Publication	Selection Bias	Study Design	Confounders	Blinding	Data Collection Methods	Withdrawals and Dropouts	Intervention Integrity	Analysis
Adedapo, A.H., et al. (2019) [27]	Weak	Moderate	Moderate	Moderate	Weak	Weak	Moderate	Moderate
Almas, K., et al. (2018) [30]	Weak	Moderate	Moderate	Moderate	Moderate	Weak	Weak	Moderate
AlSadhan, S.A. (2016) [16]	Moderate	Strong	Strong	Moderate	Strong	Moderate	Moderate	Strong
Ashwath, B., et al., (2014) [23]	Weak	Moderate	Moderate	Moderate	Moderate	Weak	Weak	Moderate
Azodo C.C., et al. (2017) [14]	Weak	Weak	Weak	Weak	Weak	Weak	Weak	Weak
Azodo C.C. (2019) [9]	Weak	Moderate	Moderate	Moderate	Moderate	Weak	Moderate	Moderate
Bhat, M.Y.S., and Alayyash, A.A. (2016) [34]	Moderate	Strong	Moderate	Moderate	Moderate	Moderate	Moderate	Moderate
Colussi, P.R.G., et al. (2017) [15]	Moderate	Strong	Moderate	Moderate	Moderate	Moderate	Moderate	Strong
Da Conceicao, M.D., et al. (2018) [19]	Moderate	Strong	Moderate	Moderate	Moderate	Moderate	Moderate	Strong
de Jongh, A., et al. (2016) [35]	Strong	Moderate	Moderate	Moderate	Moderate	Moderate	Moderate	Strong
Deolia, S.G., et al. (2018) [36]	Weak	Weak	Weak	Weak	Moderate	Weak	Weak	Moderate
Faria, S.F.S., et al. (2020) [31]	Strong	Moderate	Moderate	Moderate	Moderate	Moderate	Strong	Strong
Goel, S., et al. (2017) [29]	Weak	Weak	Weak	Weak	Moderate	Moderate	Weak	Moderate
Kim, S.Y., et al. (2015) [4]	Moderate	Moderate	Moderate	Moderate	Moderate	Weak	Moderate	Moderate
Kolo, E.S., et al. (2015) [5]	Weak	Weak	Weak	Weak	Weak	Undetermined	Weak	Weak
Kuzhalvaimozhi, P., et al. (2019) [1]	Moderate	Moderate	Weak	Moderate	Moderate	Moderate	Weak	Moderate
Mento, C. et al. (2014) [7]	Moderate	Strong	Moderate	Moderate	Moderate	Moderate	Moderate	Strong
Mrizak, J., et al. (2019) [18]	Weak	Weak	Weak	Weak	Weak	Undetermined	Weak	Weak
Mubayrik, A.B., et al. (2017) [11]	Moderate	Moderate	Moderate	Moderate	Moderate	Weak	Moderate	Moderate
Patel, J., et al. (2017) [37]	Moderate	Moderate	Moderate	Moderate	Moderate	Moderate	Weak	Moderate
Schemel-Suarez, M., et al. (2017) [2]	Weak	Weak	Weak	Weak	Weak	Undetermined	Weak	Weak
Settineri, S., et al. (2017) [24]	Moderate	Moderate	Moderate	Moderate	Moderate	Moderate	Moderate	Moderate
Settineri, S., et al. (2013) [10]	Strong	Strong	Moderate	Moderate	Strong	Moderate	Moderate	Strong
Settineri, S., et al. (2015) [6]	Moderate	Moderate	Moderate	Moderate	Moderate	Moderate	Moderate	Moderate
Settineri, S., et al. (2010) [32]	Strong	Strong	Moderate	Moderate	Strong	Moderate	Moderate	Strong
Umeizudike, K.A., et al. (2016) [38]	Weak	Weak	Weak	Weak	Weak	Weak	Undetermined	Weak
Ziaei, N., et al. (2015) [28]	Moderate	Strong	Moderate	Moderate	Strong	Moderate	Moderate	Strong

**Table 4 medicina-57-00614-t004:** Characteristics of the studies.

Publication	Aims	Sample Size and Characteristics	Group Characteristics	Mesaurement Type	Results
Adedapo, A.H., et al. (2019)	This study determines the presence of five putative periodontal pathogens, namely Actinobacillus actinomycetemcomitans, Fusobacterium nucleatum, Porphyromonas gingivalis, Prevotella intermedia and Treponema denticola, on the tongue dorsa of halitosis and nonhalitosis patients using a 16S rDNA-directed polymerase chain reaction assay.	84 patients presenting self-complaints of halitosis at the Oral Diagnosis Clinic, University College Hospital, Ibadan, between January 2008 and October 2010	The cases consisted of 24 males with mean age of 37.7 ± 10.7 years and 13 females with mean age of 40.8 ± 14.9 years. The controls consisted of 15 males with mean age of 34.9 ± 12.7 years and 22 females with mean age of 35.4 ± 14.2 years.	VSC measurements were made with a portable industrial sulfide monitor (Interscan Corp., Chatsworth, CA), zeroed on ambient air before each measurement.	Halitosis is affected by gender, with males having it more than females. Males also tend to present more with self-reported complaints of halitosis than females. Age does not appear to contribute to the incidence of halitosis.
Almas, K., et al. (2018)	Objective of the study was to evaluate self-perceived oral malodor (OM) and to correlate this with oral hygiene practices.	372 Saudi dental students: 109 students were males and 116 were females. The mean age of the subjects was 19–25.	Saudi dental students in the College of Dentistry at the Imam Abdulrahman Bin Faisal University (IAU), in the Eastern Province of Saudi Arabia.	The questionnaire contained three parts, comprising sociodemographic factors, subject’s perceptions of OM, and the social effects thereof.	The results state that there is a prevalence of malodor among dental students. In addition, regular flossing and removal of tongue coating can significantly reduce malodor.
AlSadhan, S.A. (2016)	This cross-sectional observational study was conducted to determine the prevalence of self-perceived halitosis among adults in Riyadh, Saudi Arabia, and to assess the relation of halitosis with certain sociodemographic factors, oral habits and health practices.	3000 participants, both males and females, including senior high school students, college students and employees working in governmental offices.	The college students were selected from the major universities in Riyadh. There were 15 locations for males and 15 for females (5 schools, 5 universities and 5 governmental offices for each gender).	The questionnaire was made up of 3 parts: the first part was related to certain sociodemographic factors including gender, age, educational level and employment; the second part was related to the participant’s perception of any malodor (halitosis) and its history and social effects; and the third part concerned certain oral hygiene and health habits.	The prevalence of self-perceived halitosis was 22.8% among the participants. The majority of subjects with self-perceived halitosis experienced bad breath on awakening (83.5%). Self-perceived halitosis was more prevalent among males than females, whereas no statistically significant differences were found between age groups. A statistically significant relationship was found between self-perceived halitosis and mouth cleaning time, and a significant relationship was found between self-perceived halitosis and mouth cleaning time, shisha use or cigarette smoking.
Ashwath, B., et al. (2014)	The aim of this study was to evaluate self-perception of oral malodor and oral hygiene habits amongst dental students.	285 participants in the age range of 18 to 22.	Undergraduate students of Madha Dental College and Hospital Chennai.	The questionnaire used included ten questions that assessed the presence and self-perception of halitosis and treatment (self or professional) for halitosis. It also evaluated the subjects’ oral hygiene habits, including the frequency of toothbrushing, the use of interdental aids and mouth rinsing.	The results of this study indicate a higher prevalence of halitosis among this population of dental students. The difference in reporting self-perception of halitosis between females and males was found to be statistically significant (*p* < 0.05). Significant difference was found for use of mouth wash, presence of carious teeth, bleeding gums and use of tongue cleaners between females and males.
Azodo, C.C., et al. (2017)	The objective of this study was to assess the relational impact of halitosis.		Undergraduates of University of Benin, Nigeria.	Demographic characteristics, perception of value of fresh breath in social contact, rank of halitosis in first impression impairment, frequency of encounter with halitosis sufferers and their interactional difficulties and readiness to inform sufferers about the condition were assessed using self-administered validated questionnaire.	Halitosis was considered to be the second most important factor in social interaction after body odor. The majority of respondents believed that those suffering from halitosis had difficulty finding a good job or getting married or suffered from marital disharmony.
Azodo, C.C. (2019)	The objective of this study was to determine the social trait rating of halitosis sufferers by others who are Nigerian undergraduates.	A total of 245 individuals aged between 17 and 35, comprising 100 males and 145 females.	Main (Ugbowo) campus residential undergraduates of University of Benin, Nigeria.	A questionnaire assessing health status, quality of life, intelligence, caring, trustworthiness, attractiveness, sexiness, aggressiveness, happiness, pleasantry, motivation, spirituality, satisfaction with life and social life activity of halitosis sufferers was used to collect data.	In this study, halitosis sufferers’ low ratings of pleasantness, motivation, life satisfaction and happiness made them less attractive and sexy, and this had an influence on social relationships
Bhat, M.Y.S., and Alayyash, A.A. (2016)	The goal was to try to evaluate the social stigma related to halitosis and compare it between the Saudi and British populations.	612 participants.	308 were from the Kingdom of Saudi Arabia (Jeddah and Abha) and 304 were from the United Kingdom (Cardiff, Edinburgh and Glasgow).	Questionnaire of 10 articles related to self-awareness on the state of personal halitosis and on the importance of the social embarrassment found.	Saudi and selected UK populations have experienced social embarrassment because of halitosis. A significant amount of stigma associated with halitosis persists in both countries.
Colussi, P.R.G., et al. (2017)	This study aimed to assess the impact of oral health on the quality of life of adolescents.	736 adolescents from 16 public schools and 7 private schools.	The students from public and private schools from Passo Fundo, Brazil. All students were aged between 15 and 19.	The questionnaire including demographic data, socioeconomic condition, general health behavior, health record and oral health self-perception was applied with a group of questions from the *PCATool-SB Brazil* adult version. Instrument OHIP-14 was used to assess quality of life.	Adolescents deprived of their liberty experience a strong impact on oral health quality (OHRQoL). The number of decayed teeth and exposure to smoking were associated with a greater impact on OHRQoL. Finally, tooth alignment was associated with a lower impact on OHRQoL.
Da Conceicao, M.D., et al. (2018)	The objective of the study was to try to determine the validity of the inventory of the consequences of halitosis (ICH) and to study the relationship between these consequences and SAD.	436 individuals.	Included 411 with a halitosis complaint (63.7% women; aged 18–74) and 25 without a complaint (84% women; aged 18–55). Among the 411 individuals complaining of halitosis, 164 were selected from a halitosis clinic.	Seven instruments were used for this study: a sociodemographic questionnaire, Halitosis Consequences Inventory (ICH), Social Phobia Inventory and its shortened version, the Liebowitz Social Anxiety Scale, Social Avoidance and Distress Scale and Fear of Negative Evaluation scale.	The ICH is an important tool for determining the consequences of bad breath, allowing identification of people who may require screening for social anxiety disorder (SAD).
de Jongh, A., et al. (2016)	The aim was to determine the impact of self-perceived halitosis on social interactions and the effect of using an oral rinse to manage halitosis.	For study A, 1083 subjects. For study B, 292 people.	Study A participants were members of an online survey panel assembled by the Internet survey company Panelwizard. For study B, the potential participants were personally approached in a square in the entertainment area of the city of Haarlem in the Netherlands.	The questionnaire contained questions about oral perception.	In this study, it was found that the more people perceived their oral odor, the more likely they were to maintain a distance. A subgroup of individuals was identified who reported maintaining a certain distance when meeting other people, despite a self-perceived “fresh” oral odor.
Deolia, S.G., et al. (2018)	This study aimed to understand the psychological and social effects of halitosis among young adults and to correlate their psychosocial effects with different levels of halitosis.	200 patients (male and female).	Young adults between the ages of 18 and 25 visiting a private dental college in Maharashtra who perceived themselves as sufferers of halitosis.	The questionnaire included demographics and questions related to the psychological and social impact of oral malodor on their daily lives.	The younger age group showed a higher incidence of halitosis than the older age group. Both genders did not show much difference with respect to the psychological impact of halitosis; however, the social effects of halitosis were present more in women.
Dico, G L. (2018)	The goal of this article was to discuss self-perception theory and its influence on recent research; it was argued that introspection is not an autonomous research method for discussing the epistemological implications of this behavioral attitude on psychology.				This study discussed introspection as an autonomous research method and independent source of data for psychology.
Eli, I., et al. (2001)	The aim of the study was to discuss problems involving self-perception of breath odor in patients who complain of halitosis.				The authors suggest that every patient has a breath-smelling self-image and this varies from minimal or no distortion to severe psychopathology
Faria, S.F.S., et al. (2020)	The aim of the study was to evaluate the prevalence of self-reported halitosis and its predictors and to determine the accuracy of estimates of the self-reported measures with the clinical assessment of halitosis.	5420 individuals.	Teaching staff, administrative personnel and ongoing students from Federal University of Minas Gerais.	Questionnaire containing sociodemographic, medical and dental data and self-reported halitosis measures; organoleptic test (OLT).	Self-reported halitosis was mainly associated with socioeconomic variables (age, gender, education level), oral health parameters (gingival bleeding, gingival infection, tongue coating, general oral health assessment) and impacts on daily activities (family/social environment and intimate relationships). Prevalence rates of self-reported halitosis can be considered moderate.
Goel, S., et al. (2017)	The aim of this study was to assess the level of knowledge and attitude of the Indian population toward self-perceived halitosis, about its possible causes, available treatments, its influence on social relations and level of confidence.	200 subjects participated in the study.	The outpatient department of a dental hospital.	Questionnaire that investigates sociodemographic data, presence or absence of medical conditions and habits, knowledge of the causes and treatment of bad smells and oral hygiene practices.	This exploratory study shows that Indian adolescents are aware of the problem, but there is no clear knowledge necessary to have an understanding and to adopt a healthy lifestyle.
Heboyan, A., et al. (2019)	The objective of the study was to analyze the causative factors, diagnostic methods and treatment options aimed at maintaining oral health as well as restoring an individual’s mental health, self-confidence and social status.				Early diagnosis of halitosis is crucial, and only the identification of the causal factors makes possible an adequate and customized treatment. Bad oral odor can often be influenced by general somatic diseases and the intake of various medications. Thus, treatment should be aimed at maintaining proper oral hygiene.
Kapoor, U., et al. (2016)	This article succinctly focused on the development of a systematic flow of events to arrive at the best management of the halitosis from the primary care practitioner’s point of view.				Halitosis is an extremely unpleasant feature of sociocultural interactions and can have long-term negative side effects on psychosocial relationships. An interdisciplinary approach to the treatment of halitosis should be used in order to prevent misdiagnosis or unnecessary treatment.
Kim, S.Y., et al. (2015)	This study was conducted to estimate the prevalence and associated factors of subjective halitosis in adolescents.	359,263 participants (184,801 males and 174,462 females) ranging in age from 12 to 18.	Young people from Korea.	A test was used to collect information relating to gender; age; residence; subjective health; stress level; economic level; alcohol consumption; smoking; and frequency of intake of fruit, fizzy drinks, fast food, instant pasta, sweet foods and vegetables.	In this study, factors highlighted as related to halitosis were poor health status, overweight or obesity, stress and lower economic levels. The prevalence of subjective halitosis in the adolescents studied was 23.6%. Specific psychosocial factors and dietary intake were related to halitosis.
Kisely, S. (2016)	This article discusses the two-way association between oral and mental health. On the one hand, the prospect of dental treatment can lead to anxiety and phobia. On the other, many psychiatric disorders, such as severe mental illness, affective disorders and eating disorders, are associated with dental disease.				Dental diseases can lead to teeth loss, and people with severe mental illness have 2.7 times the likelihood of losing all their teeth compared with the general population.
Kolo, E.S., et al. (2015)	This study aimed to describe the psychological and social problems of adult patients with halitosis.	36 people, 20 females and 16 males aged between 18 and 62.	Adult patients from the Aminu Kano educational hospital (northern Nigeria).	A self-administered questionnaire was explained and administered to the patients at their first visit to the clinic by the authors. The content of the questionnaire included personal bio-data and questions related to oral malodor. Subsequently, the patients were asked to assess the severity of their bad breath using the visual analog scale (VAS).	This study revealed that most of the adult patients had halitophobia, and most of them had no associated psychosocial problems. Furthermore, their bad breath was not influenced by gender or the duration of illness.
Kuzhalvaimozhi, P., et al. (2019)	Objective of the study was to assess self-perception, knowledge and attitude of halitosis among patients attending a dental hospital in Chennai.	300 patients.	Patients who visited a dental hospital in South India.	A self-structured questionnaire was implemented.	Most of the participants did not have self-perceived halitosis and most participants brushed their teeth twice a day, used mouthwash regularly and had knowledge of halitosis.
Madhushankari, G.S., et al. (2015)	This review covers the pathophysiology and various etiologies of halitosis, the knowledge of which helps improve treatment options.				The patients with halitosis initially turn to dentists for improvement of the condition. Research into volatile sulfur compounds (VSCs) and their effect on oral tissues has made the problem of halitosis a cause for real concern.
Mento, C., et al. (2014)	The aim was to determine to what extent dental anxiety can be explained by looking at patients’ characteristics solely or by considering also latent aggressiveness that could be manifested before and during the dental treatment.	A random sample of patients undergoing dental treatment at several settings located in two regions in Southern Italy.	Group 1: 153 individuals (15–26 years); Group 2: 190 individuals (27–42 years); Group 3: 170 individuals (43–70 years).	Dental Anxiety Scale (DAS), The Aggression Questionnaire (AQ), and Patient Health Questionnaire (PHQ).	The results of the sample estimates show the importance of aggressiveness in the development of dental anxiety; a significant influence is also exerted by the presence of depressive symptoms. Furthermore, there are no significant gender differences, and there is a nonlinear relationship with age; the most significant age group is between 27 and 42 years old.
Mrizak, J., et al. (2019).	Halitophobia is a condition characterized by an excessive preoccupation with the belief of having halitosis. Cognitive behavioral therapy (CBT) was successfully used to treat a man in his 20s who presented important anxiety, avoidance and safety behaviors; isolation; and depressed mood.	A 20-year-old man.	A man presenting important anxiety, avoidance and safety behaviors; isolation; and depressed mood.	Hamilton Anxiety Rating Scale (HAM-A) and Hamilton Depression Rating Scale (HAM-D).	This study suggests that CBT techniques, most commonly used in anxiety disorders and obsessive-compulsive disorder, can be adapted to halitophobia.
Mubayrik, A.B., et al. (2017)	This study was designed to measure self-perception, knowledge, and awareness of halitosis among female university students in Saudi Arabia.	440 young women.	Students from various majors in classrooms and gathering areas of King Saud University’s female campus.	The questionnaire consisted of the following: 1) a cover page with a request for cooperation and instructions; 2) demographic questions; 3) substantive questions exploring the respondents’ self-perception and awareness of halitosis along with their knowledge about causes and management of oral malodor.	The survey revealed low self-perception and limited knowledge about halitosis. Self-perception of halitosis was low, while a higher percentage indicated that they noticed people with halitosis. Most participants thought that the gastrointestinal tract was the primary source of halitosis.
Özen, M.E., et al. (2015)	This document defines subjective halitosis terminology.				In this study, it was concluded that dental practitioners’ mission is restricted to detecting whether the halitosis is objective or not, and does not include estimating, distinguishing, assessing or diagnosing psychiatric disorders.
Patel, J., et al. (2017)	The aim was to evaluate the association of social anxiety with oral hygiene status and tongue coating among patients with subjective halitosis.	321 subjects.	Subjects aged 18 or older presenting in the outpatient department with the complaint of halitosis.	Social anxiety was assessed with the 24-item self-report version of the Liebowitz Social Anxiety Scale (LSAS-SR).	This study revealed that social anxiety, poor oral hygiene and tongue coating were associated with subjective halitosis. Comparison of oral clinical parameters between the sexes revealed that poor oral hygiene was observed among male participants.
Schemel-Suárez, M., et al. (2017)	The objective of this study was to estimate the prevalence of halitosis (with subjective and objective methods), evaluate the immediate effect of chewing gum on volatile sulfur compounds (VSCs), assess the perception of halitosis by dentistry students and estimate the distribution of positive and negative frequencies, when comparing objective and subjective methods for the diagnosis of halitosis.	80 dentistry students.	The sample comprised 80 individuals: 59 women and 21 men aged from 18 to 40.	Questionnaire, the simplified oral hygiene index, the Winkel index, organoleptic test and gas chromatography.	The prevalence of halitosis in the studied sample was high, considering that it comprised healthy individuals.
Settineri, S., et al. (2017)	The study aimed to investigate the psychological impact of oral disorders on people’s emotional well-being, with a particular attention to gender and age differences.	The whole sample consisted of 263 dental patients, all belonging to private dental surgeries of the center of Messina.	For the analysis, the authors considered only the valid cases: 130 of them were females (56.8%) and 99 of them were males (43.2%), for a total of 229 participants.	The Profile of Mood States (POMS) and Oral Health Impact Profile (OHIP-14).	The results of this study showed a significant relationship between the perception of the patient’s oral health and the mood states experienced.
Settineri, S., et al. (2013)	The aim of the study was to analyze, by means of a questionnaire on disgust, any gender differences regarding the feeling of disgust in its various dimensions, viewed both individually and globally.	A sample of 1587 subjects was taken from the town of Messina (676 males and 911 females). Subjects were contacted by medicine students of Messina University Hospital.	The age range of participants was between 10 and 90.	The synthetic disgust index (SDI)	Disgust and oral contamination showed a reduction in the SDI in the group aged 18–39 years and a maximum score in the group of subjects aged 40–64 years. Disgust and oral contamination showed a reduction in the SDI in the 18–39 age group with a general tendency to decrease with age.
Settineri, S., et al. (2015)	The aim of the study was to investigate the relationship between psychosocial impact, levels of self-esteem and the possible connection with eating habits of adolescents under orthodontic treatment.	The adolescents in treatment from 0 to 60 months at the Clinic of Orthodontics and Dentistry of Messina, Messina, Italy.	Sixty-one adolescents, aged between 12 and 22 years (mean = 15.6 ± 2.8).	Eating Attitudes Test, the Rosenberg Self Esteem Scale, and the Psychosocial Impact of Dental Aesthetics Questionnaire.	The data did not show a direct connection between eating disorder and dental aesthetics. Adolescents under orthodontic treatment in the first phase of orthodontic appliance use showed peculiar eating habits and were more psychologically affected by dental aesthetics.
Settineri, S., et al. (2010)	The aim of this study was to examine behavior in a sample of Italian subjects with reference to self-reported halitosis and emotional state, specifically the presence of dental anxiety.	1052 subjects, 623 females and 388 males, aged between 15 and 65 years old.	Subjects were recruited in the waiting room of dental clinics in Messina and Reggio Calabria.	Instruments used were as follows: self-report questionnaire to detect self-reported halitosis and other variables possibly linked to it: sociodemographic data, presence or absence of medical and dental pathologies, any allergies, oral hygiene practices, medication, smoking and alcohol consumption, the importance attributed to one’s own mouth and that of others; Dental Anxiety Scale (DAS).	The rate of self-reported halitosis was 19.39%. The factors related to halitosis were anxiety about relationships with the dental patient (relational dental anxiety), alcohol consumption, gum disease, age > 30 years, female sex, poor oral hygiene, general anxiety and urinary system diseases.
Slade, P.D. (1994)	A general schematic model concerned with identifying the nature of body image is presented. This model suggests that body image may be conceived as a loose mental representation of the body that is influenced by at least 7 sets of factors.				These sets are the history of sensory input to body experience, the history of weight change/fluctuation, cultural and social norms, individual attitudes to weight and shape, cognitive and affective variables, individual psychopathology and biological variables.
Thoppay, J.R., et al. (2019)	This book, written by world authorities in the field, is a comprehensive, up-to-date guide to the specialty of oral medicine, which is concerned with the diagnosis, prevention and predominantly nonsurgical management of medically related disorders and conditions affecting the oral and maxillofacial region.				Halitosis is an unpleasant symptom that can become a social problem compromising quality of life. This condition may affect individuals of all ages and may be a transient episode or a long-lasting problem depending on the cause.
Torales, J., et al. (2017)	The aim of this brief review was to provide information on the management of oral and dental diseases that must be provided to patients with mental illness.				The most common oral and dental problems were analyzed in patients with depression, anxiety, schizophrenia or bipolar dementia, with particular attention to their management.
Umeizudike, K.A., et al. (2016)	The aim was to determine the prevalence of self-reported halitosis among dental patients observed in a university hospital.	135 dental patients.	Patients from the Oral Diagnosis/Periodontology Clinics of the Lagos University Teaching Hospital.	The self-administered questionnaires contained information on age, gender, education, marital status, ethnicity, religion and self-perception of halitosis.	The prevalence of self-reported halitosis was 14.8% and was significantly associated with age 40 years and over and male gender. About 50% perceived halitosis from themselves, 25% from family and friends and 20% from people around them. The majority (70%) of patients perceived halitosis from the mouth, 30% from the mouth and nose. The majority (75%) of subjects had perceived halitosis for more than 4 weeks.
Ziaei, N, et al. (2015)	The aim of this study was to evaluate the prevalence of halitosis and its associated factors among students.	790 high school students, aged 14 to 18; 484 were boys and 306 were girls.	High school students of Kermanshah.	The questionnaire included questions about bad breath and other associated factors (demographic information, background diseases, oral and dental problems, decay-missing-filled (DMF) index). Organoleptic evaluation was conducted.	The prevalence of halitosis in the organoleptic evaluation was 29.75% and higher in boys (32.6% male vs. 25.2% female) and 27.47% in self-perception (32.9% male vs. 19% female). The diagnostic agreement between organoleptic and self-perceived halitosis was moderate or poor.

**Table 5 medicina-57-00614-t005:** Prisma Checklist.

Section/Topic	#	Checklist Item	Reported on Page #
**title**			
Title	1	Adolescence, Adulthood and Self-Perceived Halitosis: A Role of Psychological Factors	1
**abstract**			
Structured summary	2	Background: Halitosis is a frequent condition that affects a large part of the population. Considered a “social stigma”, it can determine a number of psychological and relationship consequences that affect people’s lives. The purpose of this review is to examine the role of psychological factors in the condition of self-perceived halitosis in adolescent subjects and adulthood. Type of studies reviewed: We conducted, by the Preferred Reporting Items for Systematic Review and Meta-Analyses (PRISMA) guidelines, systematic research of the literature on PubMed and Scholar. The key terms used were halitosis, halitosis self-perception, psychological factors, breath odor and two terms related to sociorelational consequences (“Halitosis and Social Relationship” OR “Social Issue of Halitosis”). Initial research identified 3008 articles. As a result of the inclusion and exclusion criteria, the number of publications was reduced to 38. Results: According to the literature examined, halitosis is a condition that is rarely self-perceived. In general, women have a greater ability to recognize it than men. Several factors can affect the perception of the dental condition, such as socioeconomic status, emotional state and body image. Conclusion and practical implication: Self-perceived halitosis could have a significant impact on the patient’s quality of life. Among the most frequent consequences are found anxiety, reduced levels of self-esteem, misinterpretation of other people’s attitudes and embarrassment and relational discomfort that often result in social isolation.	1
**introduction**			
Rationale	3	Halitosis, commonly called bad breath, is a problem that can affect both the external, relational, social communication and internal, psychological sphere, with implications on perceived quality of life [1]. People are usually unaware of their breath, and when they become aware of it, they incorrectly attribute the cause of their condition. Halitosis is a frequent condition, present in 50–65% of the world population [2]. Although it is a significant source of discomfort, a precise estimation of prevalence is not possible because epidemiological studies are limited. This limitation could be determined by several factors, such as the absence of a standardized method for assessment of the disease, the difficulty in recognizing its presence and the likelihood that it is in some cases transient, which is why it is underreported in epidemiological data [3]. The problem of people with halitosis is that this condition can often remain unnoticed because people are generally unaware of the quality of their oral odor. Considered by Kolo [5] as a “social stigma”, it could become an obsession that dominates a person’s life, determining the onset of factors such as anxiety and psychosocial stress. One theme of studies on self-perceived halitosis shows that the problem is often not self-perceived [20,21]. Self-reported halitosis tends to be underestimated mainly because individuals may have difficulty in detecting their own smell or feel embarrassed to expose themselves, in line with the intimate dimension of the relationship with their mouth [22]. Subjects with a good body image pay more attention to their mouth and oral malodor [23]. Recent studies confirm that self-perception of halitosis may be related to a psychogenic or psychosomatic disorder and has a strong psychological impact [31,32].	2,3
Objectives	4	The aim of this work is to examine the role of psychological factors in the condition of self-perceived halitosis in adolescent subjects and adulthood.	1
**methods**			
Protocol and registration	5	We used the following search terms: halitosis, self-perception halitosis, psychological factors, breath odor and two terms related to sociorelational consequences (“Halitosis and Social Relationship” OR “ Social Issue of Halitosis”). In the initial search, we identified 3008 publications; after applying inclusion and exclusion criteria, we analyzed 38 studies.	4,5
Eligibility criteria	6	We used the following eligibility criteria: English language, publication in peer-reviewed scientific journals, quantitative information about self-perceived halitosis and oral hygiene. Articles were excluded based on title and abstract screen; article type of review article, editorial comment or case report; and year of publication prior to 2010.	4,5
Information sources	7	This review was conducted according to Preferred Reporting Items for Systematic Reviews and Meta-Analyses [33]. To identify the studies, we performed a systematic literature search on the PubMed, Scholar and ScienceDirect databases. The search was limited to studies written in English. We identified the literature from January 2010 to January 2020 using four key terms related to self-perceived halitosis, namely halitosis, self-perception halitosis, psychological factors and breath odor, and two terms related to sociorelational consequences (“Halitosis and Social Relationship” OR “ Social Issue of Halitosis”). The articles from 1994 and 2001 were only cited to introduce the body image theory and to link it to the perception that each individual has of his or her own smell.	4,5
Search	8	Articles were selected by title and abstract; the entire article was read if the title and abstract were related to the specific issue of adolescence, adulthood and self-perceived halitosis and if the article potentially met the inclusion criteria. References of the selected articles were also examined in order to identify additional studies meeting the inclusion criteria.	4,5
Study selection	9	We performed a systematic literature search on the PubMed, Scholar and ScienceDirect databases. The search was limited to studies written in English. We identified the literature from January 2010 to January 2020 using four key terms related to self-perceived halitosis, namely halitosis, self-perception halitosis, psychological factors and breath odor, and two terms related to sociorelational consequences (“Halitosis and Social Relationship” OR “ Social Issue of Halitosis”).	5
Data collection process	10	Articles were selected online in relation to the title and abstracts; articles were read in full when titles and abstracts were consistent with the objective of our study. Following this procedure, we found 889 articles on the PubMed database, 1950 articles on Scholar and 169 articles on ScienceDirect; after applying the inclusion and exclusion criteria, the total number of relevant publications was reduced to 38.	5
Data items	11	Not specified.	
Risk of bias in individual studies	12	Across the included studies in this review, a potential database bias should be considered. Only articles written in English language were used, which might have compromised access to articles published in other languages.	
Summary measures	13	The list of search terms entered into the PubMed, Scholar and ScienceDirect search is described in Table 1.	Table 1
Synthesis of results	14	The search in the PubMed, Scholar and ScienceDirect databases provided a total of 3008 articles; no further studies meeting the inclusion criteria were identified. After eliminating duplicates, further studies were excluded according to the inclusion and exclusion criteria. After screening, 38 studies were selected as appropriate for the present review.	5

## Data Availability

The study did not report any data.

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
