# Peer review of "Adolescence, Adulthood and Self-Perceived Halitosis: A Role of Psychological Factors"

_medicina, 2021, doi:10.3390/medicina57060614_

Round 1
Reviewer 1 Report
Line 30 : Please control the terms « self perceived » and « auto-perceived » : there meaning is similar ; you wish to conserv both of them ? in that case, may you explain why ?
Lines 108-109 : is there really a statistically difference between males and females with such closed scores ? What is the biblio reference ? (usually, there is no difference between gender…) In the column results of the table, following the references, we can observe both
Line 119 : « hostile to going to the dentist… » not well-turned
P6, Ref Almas, column resuts : « regular flossing » twice
Ref Heboyan, column resuts : « oral hygiene » twice
Ref Kuzhalvai, column resuts : « Most of the…
Ref Ozen, column resuts : « Practitioners.…
Ref Schemel, column resuts : « was high…,
Author Response
Comments about the Referee 1:
We thank the Editor and the Reviewer for giving us the opportunity to review and improve the quality of the manuscript.
1.Comment about Line 30: “Please control the terms « self-perceived » and « auto-perceived »: there meaning is similar; you wish to conserv both of them ? in that case, may you explain why?”
We corrected this part and have chosen to use the term halitosis-self-perceived
- Comment about Lines 108-109: “is there really a statistically difference between males and females with such closed scores? What is the biblio reference? (usually, there is no difference between gender…) In the column results of the table, following the references, we can observe both.”
The reference of the article is “Ashwath, B.; Vijayalakshmi, R.; Malini, S., Self-perceived halitosis and oral hygiene habits among undergraduate dental students. Journal of Indian Society of Perio-dontology 2014, 18 (3), 357”.
We checked and corrected the values that were reported in the abstract. In the results section was reported: “Among males, 26 (21.7%) reported perceiving halitosis, 77 (64.2%) gave a negative answer, and 17 (14.2%) were not aware of its presence or absence; where in females, 49 (35.3%) reported self-perception, 51 (36.7%) gave a negative response, and 39 (28.1%) were not aware of its presence or absence. The difference was found to be significant (P < 0.05)”
- Comment about Line 119: “hostile to going to the dentist… » not well-turned
We corrected this sentence.
- Comment about P6:
- Ref Almas, column resuts: « regular flossing » twice
- Ref Heboyan, column resuts: « oral hygiene » twice
- Ref Kuzhalvai, column resuts: « Most of the…
- Ref Ozen, column resuts: « Practitioners.…
- Ref Schemel, column resuts: « was high…,
We corrected this part.
Reviewer 2 Report
This is an interesting study however the authors have not addressed my primary concern that the methodology is unclear and incomplete. Specifically: 1. the PICO is not explicit, 2. there is no quality assessment of the papers included and therefore it is not possible to determine the quality of the evidence, and 3. I don't understand how the authors have analysed the papers (What is a systematic analysis? I suspect they mean a thematic analysis).
A thematic analysis will help to group the papers into themes eg. self-perception of halitosis; gender differences; halitosis paradox; etc This will make your results for more meaningful to the reader and help shape the discussion.
My concerns about the use of strong language eg line 119 and the statistical significance in line 110 remain.
It would take very little additional work for the authors to add a quality assessment of the included papers (eg EPHPP tool for quantitative studies and CASP for qualitative) to their methods and results. The heterogeneity of the studies means that meta analysis is not possible - and that's ok - but an assessment of the included studies is an important inclusion in a systematic review.
Author Response
Comment about the Referee 2:
This is an interesting study however the authors have not addressed my primary concern that the methodology is unclear and incomplete.
1.Comment about the PICO:” the PICO is not explicit.”
We have revised and added more informations to clarify the PICO.
|
PICOS Detail Component |
|
|
Partecipants |
The participants in the study were healthy subjects. |
|
Interventions |
Participants in some studies underwent organoleptic tests to measure breathing odor and specific questionnaires were given |
|
Comparisons |
Some patients with halitosis were compared with healthy subjects |
|
Outcomes |
This study has identified the importance of the consequences that the perception of bad breath has on the psycho-relational side and, in general, on patients’ quality of life |
|
Study designs |
Observational studies |
- Comment about the quality assessment: “there is no quality assessment of the papers included and therefore it is not possible to determine the quality of the evidence.” “It would take very little additional work for the authors to add a quality assessment of the included papers (eg EPHPP tool for quantitative studies and CASP for qualitative) to their methods and results. The heterogeneity of the studies means that meta-analysis is not possible - and that's ok - but an assessment of the included studies is an important inclusion in a systematic review.”
Thank you for the valuable advice. We have now added the quality assessment of the included documents. As suggested, we used CASP tool for qualitative studies and EPHPP tool for quantitative studies. Most of the qualitative studies were found to have good quality, while some of the quantitative works showed relevant methodological weaknesses such as selection bias, unsatisfactory design and presence of confounders. We have now added results of CASP and EPHPP quality assessment in Table 2 and 3 respectively.
- Comment about analysis the paper: “I don't understand how the authors have analysed the papers (What is a systematic analysis? I suspect they mean a thematic analysis).”
Thank you for your kind attention. We have analysed the papers through a thematic analysis.
- 4. Comment about line 119 and 110: “My concerns about the use of strong language eg line 119 and the statistical significance in line 110 remain.”
- We corrected the language of line 119.
- We checked and corrected the values that were reported in the abstract. In the results is reported: “Among males, 26 (21.7%) reported perceiving halitosis, 77 (64.2%) gave a negative answer, and 17 (14.2%) were not aware of its presence or absence; where in females, 49 (35.3%) reported self-perception, 51 (36.7%) gave a negative response, and 39 (28.1%) were not aware of its presence or absence. The difference was found to be significant (P < 0.05)”.
Thank you very much for your attention and the Reviewers’ evaluation and comments on our manuscript: again, we appreciate all your insightful comments, and we tried to be responsive to them.
Thank you for taking the time to help us to revise and improve our manuscript.
We look forward to hearing from you at your earliest convenience.
Best Regards,
Carmela Mento
Round 2
Reviewer 2 Report
This paper is an important contribution to the literature. The changes you have made have significantly enhanced the quality of the manuscript and the significance of your findings. Congratulations.
This manuscript is a resubmission of an earlier submission. The following is a list of the peer review reports and author responses from that submission.
Round 1
Reviewer 1 Report
Thank you for the opportunity to review this potentially important study about halitosis perceptions and impacts. This is an interesting topic. Unfortunately there are some issues with the method and the write up. My comments are separated into minor and major issues.
Minor:
- There is incorrect use of the dash (-) eg lines 18, 25, 33, 41 59, 61, 73, 70, etc
- The entire manuscript needs to be reviewed for correct us of English language see eg line 17
- There is some very strong language used - perhaps use less emotive/loaded terms eg line 116
- Are you sure about line 108 'In the perception of halitosis, the gender difference is statistically significant: it was reported by 44.1% of
109 males and 45.32% of females.' - this doesn't look statistically significant to me??
Major:
- The methodology is very weak. There is no quality assessment of the papers (eg EPHPP or CASP or any other tool) and so the reader has no idea how reliable the findings are
- As a result of 1, bias is not properly addressed
- Whilst the search strategy is detailed, the remainder of the methods is unclear and as a result the reader does not know how the conclusions were drawn. I assume a thematic analysis was undertaken?
- The methodology should include the PICO as well as the method/s for data analysis.
Reviewer 2 Report
This article is a very interesting review of a topic fewly discussed in the litterature. The reading is nice ; the introduction and materiel & method parts are well documented and described.
Nevertheless, three observations can be underlined:
1/ Even if "halitosis paradox" has been cited (line 102), we don't feel the complexity , the absence of relationship between self-halitosis and the genuine halitosis. This is of importance as "Self perceived halitosis" represents the core of this paper
2/ PRISMA's items are consisting often in more than 14 items; the following ones are considering the Results chapter, the data and outcomes. It would have been particulary useful to get a table with those informations for each study reported in table 3.
3/ Considering the bias cited in part 2.3, consisting in the choice of english papers: a comment about the presence of observationel studies, and the absence of any RCTs would have been of interest.